# An Iterative Algorithm for Predicting Seafloor Topography from Gravity Anomalies

Jinhai Yu [1], Bang An [1,*], Huan Xu [1], Zhongmiao Sun [2], Yuwei Tian [1] and Qiuyu Wang [1]

1 Key Laboratory of Computational Geodynamics, College of Earth and Planetary Sciences, University of Chinese Academy of Sciences, Beijing 100049, China
2 National Key Laboratory of Geoinformation Engineering, Xi'an 710054, China
* Correspondence: anbang19@mails.ucas.ac.cn

**Abstract:** As high-resolution global coverage cannot easily be achieved by direct bathymetry, the use of gravity data is an alternative method to predict seafloor topography. Currently, the commonly used algorithms for predicting seafloor topography are mainly based on the approximate linear relationship between topography and gravity anomaly. In actual application, it is also necessary to process the corresponding data according to some empirical methods, which can cause uncertainty in predicting topography. In this paper, we established analytical observation equations between the gravity anomaly and topography, and obtained the corresponding iterative solving method based on the least square method after linearizing the equations. Furthermore, the regularization method and piecewise bilinear interpolation function are introduced into the observation equations to effectively suppress the high-frequency effect of the boundary sea region and the low-frequency effect of the far sea region. Finally, the seafloor topography beneath a sea region (117.25°–118.25°E, 13.85°–14.85°N) in the South China Sea is predicted as an actual application, where gravity anomaly data of the study area with a resolution of $1' \times 1'$ are from the DTU17 model. Comparing the prediction results with the data of ship soundings from the National Geophysical Data Center (NGDC), the root-mean-square (RMS) error and relative error can be up to 127.4 m and approximately 3.4%, respectively.

**Keywords:** gravity anomalies; topography; regularization; bilinear interpolation





## 1. Introduction

As a natural density interface of the earth, the seafloor topography plays an important role in many geoscience fields [1–6]. Apart from the direct measurement of sea depth by single/multi-beam technology, remote sensing technology and marine gravity data are also important for the indirect measurement of sea depth [7–9]. For a multi-beam echosounder, although it has a high accuracy, the distribution of actual ship soundings data is very sparse due to large time consumption and high cost [10]. For remote sensing technology, satellites including Sentinel-2, ICEsat, and others can measure the depth of shallow seas near islands and reefs with an accuracy of less than 1 m. However, depth prediction by remote sensing technology is limited as it can only capture the topographic features of sea areas with depths less than 18 m [11]. Compared with multi-beam echosounder and remote sensing technology, marine gravity data are well distributed across the ocean. For example, geoid heights with a resolution better than 2 km can be obtained by integrating data of many altimetry satellites, such as Topex/Poseion (T/P) satellite, Jason-1 satellite, and Cryosat-2 satellite, etc. [12], from which gravity data, such as the gravity anomaly and gravity gradients, can be satisfactorily computed [13,14]. Therefore, a highly effective method for mapping seafloor topography is as follows: first, gravity data with a high resolution (e.g., 2 km) can be used to predict seafloor topography with the same resolution, and the data of sea depths from ship soundings and remote sensing can then be combined to refine the topography. In fact, using gravity anomaly to predict seafloor topography

can effectively fill the lack of ship sounding data and improve the overall accuracy of the seafloor topography.

Considering the research status quo of using gravity data to predict seafloor topography, gravity anomaly has been used as the main type of data [15–17]. Additionally, vertical gravity gradient data have also been used to a lesser extent [18–21]. The prediction methods are mainly divided into the spatial- and frequency-domain methods. A typical representative of the spatial method is the gravity-geologic method (GGM) based on the Bouguer correction formula; namely, the relationship between the gravity generated by an infinite uniform thick plate and the height of the thick plate is linear; thus, the relationship between the gravity anomaly and seafloor topography can be fitted by existing ship soundings [22,23]. Essentially, the GGM is a fitting method that can be easily computed, but it cannot predict heavily undulating seafloor topography [20]. The frequency-domain method for predicting seafloor topography is based on the Parker formula [24] that is essentially a first-order approximate formula omitting the high-order terms of sea depth. The frequency-domain method needs to consider the flexural isostatic compensation theory to improve its accuracy; thus, more geophysical parameters are required [25,26]. Although the Parker formula has been widely used in predicting seafloor topography, the omitted higher-order terms still have a large impact in rugged areas. Yang et al. [20] pointed out that the Parker formula is less accurate in sea areas with large variations in seafloor topography.

The purpose of this paper is to directly compute the gravity generated by a rectangular prism to establish a rigorous set of observation equations between the gravity anomaly and sea depth. Subsequently, the solvability and anti-error properties of the observation equations are investigated by numerical simulation. Simultaneously, the spectral characteristics of the measured gravity anomaly are also analyzed to eliminate disturbances in the gravity anomaly and accurately predict seafloor topography. Finally, to verify the effectiveness of our algorithm, a sea region in the South China Sea is selected as a test area to predict its seafloor topography.

## 2. Theory and Methods

### 2.1. Computational Formula of Gravity Generated by a Prism

The calculation of gravity anomalies is based on the work of Nagy et al. [27,28] and Blakely [29], who derived a rigorous mathematical expression for the gravity generated by a single rectangular column of constant density using an integral method, while complex structures can be combined and superimposed by a series of rectangular columns, thus allowing the forward calculation of gravity anomalies generated by complex terrain.

The mathematical expression for the vertical gravity generated by a rectangular prism of constant density is introduced here. We assumed that $A = \{(\xi, \eta, \zeta); x_1 \leq \xi \leq x_2, y_1 \leq \eta \leq y_2, z_1 \leq \zeta \leq z_2\}$ is a rectangular prism of constant density $\rho_A$ in coordinates $O\text{-}\xi\eta\zeta$, and $Q(x_Q, y_Q, z_Q)$ is a point outside $A$. Introducing the notations

$$
\begin{cases}
\xi_1 = x_1 - x_Q, & \xi_2 = x_2 - x_Q \\
\eta_1 = y_1 - y_Q, & \eta_2 = y_2 - y_Q \\
\zeta_1 = z_1 - z_Q, & \zeta_2 = z_2 - z_Q
\end{cases}
\tag{1}
$$

and

$$
r = \sqrt{(\xi - x_Q)^2 + (\eta - y_Q)^2 + (\zeta - z_Q)^2}
\tag{2}
$$

then the vertical gravity at point $Q$ generated by $A$ is

$$
\begin{aligned}
g_A(x_Q, y_Q, z_Q) &= G\rho_A \iiint_A \frac{\zeta - z_Q}{\sqrt{[(\xi - x_Q)^2 + (\eta - y_Q)^2 + (\zeta - z_Q)^2]^3}} \, d\xi \, d\eta \, d\zeta \\
&= G\rho_A \left[ \left| \left| \left| \xi \ln(\eta + r) + \eta \ln(\xi + r) - \zeta \arctan \frac{\xi\eta}{\zeta r} \right|_{\xi_1}^{\xi_2} \right|_{\eta_1}^{\eta_2} \right|_{\zeta_1}^{\zeta_2} \right]
\end{aligned}
\tag{3}
$$

where the vertical gravity represents the derivation of the gravitational potential with respect to the variable $z$.

If $z_1 = 0$ in the prism $A$ and $z_Q = 0$, assuming that $R = \{(x,y); x_1 \le x \le x_2, y_1 \le y \le y_2\}$ is the rectangular region corresponding to the prism $A$ on the sea surface, then Equation (3) can be simplified as

$$g_A(x_Q, y_Q, 0) = G\rho_A \cdot J_R(x_Q, y_Q, z_2) \tag{4}$$

where $z_2$ is the sea depth of $A$, and

$$J_R(x_Q, y_Q, z_2) = \left\| \left| \xi \ln \frac{\eta + \sqrt{\xi^2 + \eta^2 + z_2^2}}{\eta + \sqrt{\xi^2 + \eta^2}} + \eta \ln \frac{\xi + \sqrt{\xi^2 + \eta^2 + z_2^2}}{\xi + \sqrt{\xi^2 + \eta^2}} - z_2 \arctan \frac{\xi\eta}{z_2\sqrt{\xi^2 + \eta^2 + z_2^2}} \right|_{\xi_1}^{\xi_2} \right|_{\eta_1}^{\eta_2} \tag{5}$$

Notably, $(x_Q, y_Q)$ in Equations (4) and (5) can assume the range of whole local sea surface $O - xy$. Therefore, Equations (4) and (5) are analytical formulas for the gravity on the sea surface generated by the rectangular prism $A = \{(\xi, \eta, \zeta); x_1 \le \xi \le x_2, y_1 \le \eta \le y_2, z_1 \le \zeta \le z_2\}$ below the sea surface. The derivation details and expression form refer to the work of Nagy et al. [28]

### 2.2. Establishment of the Observation Equations for Sea Depth from the Gravity Anomaly

2.2.1. Observation Equations Only for the Target Area

A local coordinate system $O - xyz$ is established for the target area $R$ by considering the local sea surface as $O - xy$ and the $z$-axis downward (away from the sea surface) (Shown in Figure 1). Assuming that $R = \{(x,y); -a \le x \le a, -a \le y \le a\}$ is a square area on the sea surface (called the target area), $h(x,y)$ is the sea depth at $(x,y)$ (to be solved), and $\Omega = \{(x,y,z); (x,y) \in R, 0 \le z \le h(x,y)\}$ is the curved column formed by the region of seawater below $R$. If seawater in $\Omega$ is replaced by rocks beneath the seafloor, then the gravity anomaly at point $Q(x_Q, y_Q)$ on $R$ generated by $\Omega$ is

$$\delta g_R(x_Q, y_Q) = G\Delta\rho \iiint_\Omega \frac{\zeta}{\sqrt{[(\xi - x_Q)^2 + (\eta - y_Q)^2 + \zeta^2]}^3} d\xi d\eta d\zeta \tag{6}$$

where $\Delta\rho = \rho_w - \rho_c$, and $\rho_c$ and $\rho_w$ are the average densities of the lithosphere and seawater, respectively. Assuming that $t$ is the step length, and $(x_i, y_j)$ is the partition points of $R$, wherein $x_i = i \cdot t$, $y_j = j \cdot t$ and $a = N \cdot t$. If the length $t$ is small, the curved column below the segmented subdomain $R_{ij} = [x_i, x_{i+1}] \times [y_j, y_{j+1}]$ of $R$ can then be approximated as a prism $\Omega_{ij} = \{x_i \le x \le x_{i+1}, y_j \le y \le y_{j+1}, 0 \le z \le h_{ij}\}$, where $h_{ij}$ is the average depth of $[x_i, x_{i+1}] \times [y_j, y_{j+1}]$. Using Equation (4), Equation (6) can be expressed as

$$\delta g_R(x_Q, y_Q) = G \cdot \Delta\rho \sum_{i,j=-N}^{N-1} J_{R_{ij}}(x_Q, y_Q, h_{ij}) \tag{7}$$

where $J_{R_{ij}}$ is computed by Equation (6). If the gravity anomaly $\delta g_R(x_Q, y_Q)$ on $R$ is obtained in advance, then Equation (7) is the set of observation equations for sea depths $h_{ij}$.

Parker [24] derived a linear formula between Fourier transforms of $g_R(x,y)$ and $h(x,y)$ from Equation (6) after omitting high-order quantities of $h(x,y)$ [5,30]. As the frequency-domain method, this formula is widely used in predicting seafloor topography. Compared with the Parker formula, Equation (7) obviously has higher accuracy, since it is derived without any approximation. In fact, the omitted high-order quantity $O(f \cdot h^2)$ in the Parker formula still has a large impact on high frequency $f$ and large sea depth $h$.

Variations in the gravity anomaly on the sea surface are mainly caused by the mass deficit of the seafloor topography, the density anomaly of the lithosphere, and the isostatic compensation of mass below the lithosphere. The mass deficit by seafloor topography sig-

nificantly contributes to the gravity anomaly on the sea surface, whereas the contributions of other factors are smoothed by upward continuation [31].

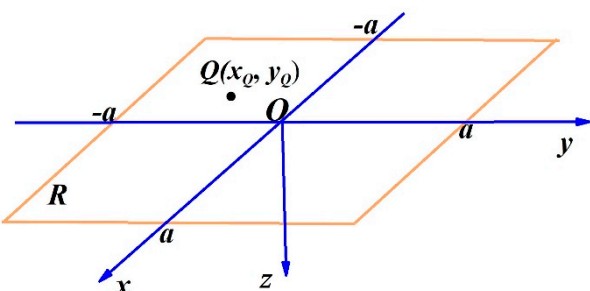

**Figure 1.** A local coordinate system is established for the target area, where *x*-axis points north, *y*-axis points east and *z*-axis points downward, while *Q* is a point in *R*.

In terms of the magnitudes of the influences, the closer the distance to *R*, the larger the influence on the gravity anomaly on *R*. In the following Figure 2, the regions that have an effect on the gravity anomaly on *R* are divided into boundary, far, and deep regions, and the methods to deal with these effects are individually investigated.

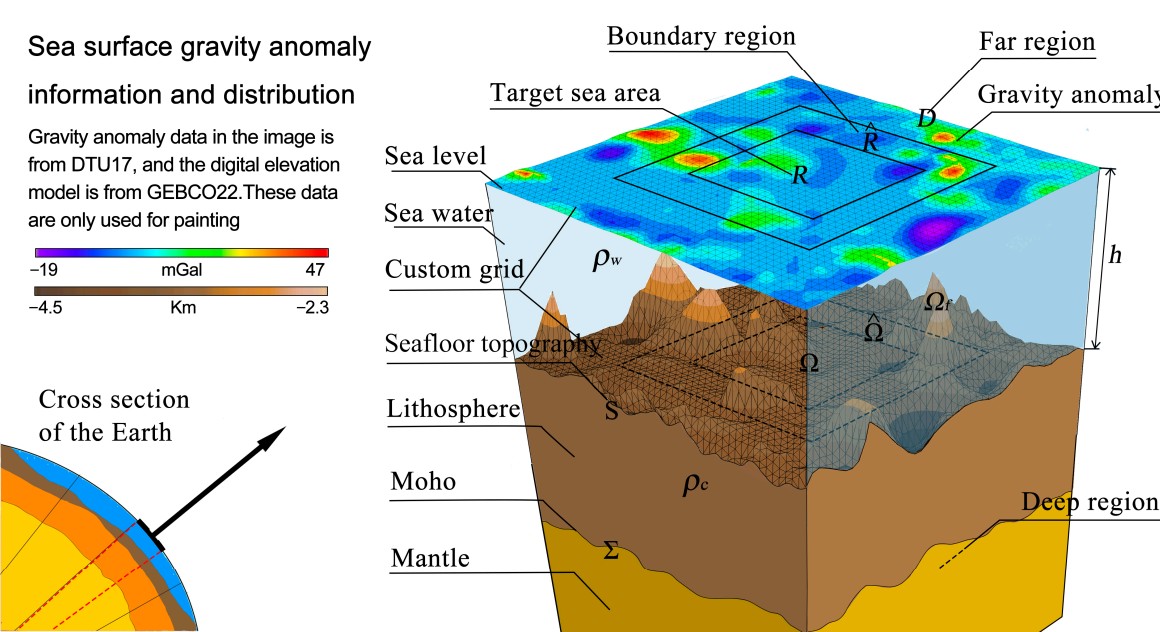

**Figure 2.** Information and distribution of the gravity anomaly on the sea surface, where $\Sigma$ and *S* represent the Moho surface and seafloor topography, respectively; *R*, $\hat{R}$ and *D* represent the corresponding target, boundary and far regions on the sea surface, respectively; $\rho_w$ and $\rho_c$ represent the densities of seawater and bedrock, respectively.

### 2.2.2. Observation Equations by Considering the Boundary Region

By extending *R* outside the boundary by *M* steps, a region $\hat{R} = \{(x, y); -(M + N)t \leq x, y \leq (M + N)t\}$ is introduced. Subsequently, $\hat{R} - R$ is called the boundary region of *R*, and the effect of its topography on solving the sea depth below *R* is called the boundary effect. By considering the boundary effect, Equation (7) can be written as

$$G\Delta\rho \sum_{i,j=-(N+M)}^{N+M-1} J_{R_{ij}}(x_Q, y_Q, h_{ij}) = \delta g_{\hat{R}}(x_Q, y_Q), \quad (x_Q, y_Q) \in R \qquad (8)$$

where $\delta g_{\hat{R}}(x_Q, y_Q)$ is the gravity anomaly generated by the curved column $\hat{\Omega}$ formed by the seawater below $\hat{R}$. Equation (8) is the system of observation equations for sea depth $h_{ij}$ below $R$ after considering the boundary effect.

We then subdivided the grid points on $R$, namely, we consider $(\hat{x}_p, \hat{y}_q) \in R$, where $\hat{x}_p = \frac{pt}{2}, \hat{y}_q = \frac{qt}{2}$, and $p, q = 0, \pm 1, \cdots, \pm 2N$. If the gravity anomaly $\delta g_{\hat{R}}(\hat{x}_p, \hat{y}_q)$ is known, the following equation is obtained

$$G\Delta\rho \sum_{i,j=-(N+M)}^{N+M-1} J_{R_{ij}}(\hat{x}_p, \hat{y}_q, h_{ij}) = \delta g_{\hat{R}}(\hat{x}_p, \hat{y}_q) \tag{9}$$

where $p, q = 0, \pm 1, \cdots, \pm 2N$. Notably, the number of equations in Equation (9) is $(4N + 1)^2$ and the number of unknowns is $(2N + 2M)^2$; thus, $N \geq M$ is required to ensure that Equation (9) has enough equations. As Equation (9) is nonlinear with respect to the solved variables $h_{ij}$, linearization must be conducted. After linearization, the corresponding iterative procedure for $h_{ij}$ is

$$G\Delta\rho \sum_{i,j=-(N+M)}^{N+M-1} \frac{\partial J_{R_{ij}}(\hat{x}_p, \hat{y}_q, h_{ij}^{(k)})}{\partial h_{ij}^{(k)}}[h_{ij}^{(k+1)} - h_{ij}^{(k)}] = \delta g_{\hat{R}}(\hat{x}_p, \hat{y}_q) - G\Delta\rho \sum_{i,j=-(N+M)}^{N+M-1} J_{R_{ij}}(\hat{x}_p, \hat{y}_q, h_{ij}^{(k)}) \tag{10}$$

where $k = 0, 1, \cdots$, and $h_{ij}^{(0)}$ is the iterative initial value of $h_{ij}$.

### 2.2.3. Effect of the Deep Region: Correction for the Moho Undulation

Figure 2 shows that the effect of the deep region of Earth on the gravity anomaly on $R$ is mainly derived from the undulation of the Moho surface; hereafter, this effect is simply called the "deep effect." In physical geodesy, Vening-Meinesz or the Airy isostatic theory is usually used to determine the Moho surface. In this paper, the Airy isostatic theory is recommended. Notably, for a seamount with depth $h$, if $\rho_w$, $\rho_c$ and $\rho_m$ are the densities of seawater, lithosphere, and upper mantle, respectively, and $L$ is the height of the Moho surface uplift corresponding to the seamount, then $L = \frac{\rho_c - \rho_w}{\rho_m - \rho_c} h$ and $T_0 - L$ represent the depth of the Moho surface from the sea surface below the seamount [32,33], where $T_0 = 25$ km is usually chosen.

According to the Airy isostatic theory, the depth $T_0 - L$ of the Moho surface can be directly derived from the depth $h$ of the seamount. As the depth of the Moho surface from sea surface is much larger than the depth $h$ of the seamount, the effect of the Moho surface undulation on the gravity anomaly can be easily reduced with the help of the Airy isostatic theory after the seafloor topography is preliminarily solved. Therefore, the deep effect, such as the Moho surface undulation, can be corrected in advance.

### 2.2.4. System of Observation Equations in the General Case

If the target area $R$ is extended to whole sea surface $S$ in Equation (7), and the gravity anomaly generated by the density difference of seawater with respect to the lithosphere is $\delta_S g$, then considering $(x, y) \in R$, we have

$$G\Delta\rho \sum_{i,j=-(N+M)}^{N+M-1} J_{R_{ij}}(x, y, h_{ij}) = \delta g_S(x, y) - \delta g_D(x, y) \tag{11}$$

where $D$ is the far region (Figure 2) and $\delta_D g$ is the gravity anomaly generated by the density difference of seawater with respect to the lithosphere below $D$. Notably, $\delta_D g$ is the effect of the far region on the gravity anomaly and is simply called the far effect hereafter.

Generally, assuming that $v$ is the Earth's gravitational potential and $v_S$ is the gravitational potential generated by replacing seawater in the ocean with the rock in the

lithosphere, we obtain $\delta g_S = \frac{\partial(v-v_S)}{\partial z}$ on $R$. If $V$ is the Somigliana gravitational potential and $T = v - V$ is the disturbing potential, we obtain the following on $R$

$$\frac{\partial T}{\partial z} = \delta g_S + \frac{\partial(v_S - V)}{\partial z} \tag{12}$$

If the isostatic theory is used to eliminate the effect of the Moho surface, then the deep effect $\frac{\partial(v_S - V)}{\partial z}$ exhibits characteristics of long waves on the sea surface according to the circle construction of Earth density (i.e., Earth density is distributed in a laminar pattern). Substituting Equation (12) into Equation (11), then at $(x, y) \in R$, we have

$$G\Delta\rho \sum_{i,j=-(N+M)}^{N+M-1} J_{R_{ij}}(x, y, h_{ij}) = \frac{\partial T(x,y)}{\partial z} - \delta g_D(x,y) - \frac{\partial(v_S - V)}{\partial z} \tag{13}$$

As the coordinate system $O$-$xyz$ (Figure 1) is locally established near $R$, $\frac{\partial T}{\partial z} = -\frac{\partial T}{\partial r} = \delta g$ on $R$, where $\delta g$ is the gravity anomaly based on the Somigliana gravity field, for which data can be obtained by the known gravity field models, such as EGM2008 or DTU17. Assuming that $F(x, y) = -\delta g_D(x, y) - \frac{\partial(v_S - V)}{\partial z}$, then at $(x, y) \in R$, we have

$$G\Delta\rho \sum_{i,j=-(N+M)}^{N+M-1} J_{\Omega_{ij}}(x, y, h_{ij}) = \delta g(x, y) + F(x, y) \tag{14}$$

where $F(x, y)$ is the long wave (or low frequency) on $R$ and is continuous. If the values of $F_{ij} = F(x_i, y_j)$ at partition points of $R$ are known, then a bilinear interpolation function $\hat{F}_{ij}(x, y)$ can be obtained using the function values $F_{ij}$, $F_{i+1,j}$, $F_{i,j+1}$, and $F_{i+1,j+1}$ on each sub rectangle of $R_{ij}$. In general, for any $(x, y) \in R$, we assumed that $\hat{F}(x, y, \mathbf{F}) = \hat{F}_{ij}(x, y)$, where $(x, y) \in R_{ij}$ and $\mathbf{F}$ is the vector comprising values $F_{ij}$ at all partition points. Notably, $\hat{F}(x, y, \mathbf{F})$ is continuous on $R$ with respect to $(x, y)$ and linear with respect to $\mathbf{F}$. Moreover, $\hat{F}(x, y, \mathbf{F})$ is the piecewise bilinear interpolation function of $F(x, y)$. $F(x, y)$ is the long wave on $R$ and its wavelength is much larger than the step length $t$ to partition $R$, so $\hat{F}(x, y, \mathbf{F}) \approx F(x, y)$. Thus, Equation (14) can be finally expressed as

$$G\Delta\rho \sum_{i,j=-(N+M)}^{N+M-1} J_{\Omega_{ij}}(x, y, h_{ij}) = \delta g(x, y) + \hat{F}(x, y, \mathbf{F}), \quad (x, y) \in R \tag{15}$$

where $h_{ij}$ and $F_{ij}$ are the variables to be solved.

So far, we have established three sets of observation equations for predicting sea depth $h_{ij}$, namely, Equations (7), (9), and (15), where Equation (7) is established by only considering the target region $R$; Equation (9) is established after considering the boundary effect of $R$; and Equation (15) is established after considering both the boundary effect of $R$ and the far effect. As the observation equations are nonlinear with respect to the sea depth $h_{ij}$, they should be linearized for $h_{ij}$ in actual computation. For example, Equation (10) is the result of the linearization of Equation (9). Additionally, Equation (15) is linear with respect to the variable F; thus, only the variable $h_{ij}$ should be linearized in Equation (15).

### 2.3. Regularization Method for the Solving Equations

This paragraph mainly discusses the solvability problem for observation equations. To ensure that the descriptions are clear, only Equation (10) is discussed as an example. Introducing matrix

$$A_k = G\Delta\rho \left( \frac{\partial J_{R_{ij}}(\hat{x}_p, \hat{y}_q, h_{ij}^{(k)})}{\partial h_{ij}^{(k)}} \right)_{pq,ij} \tag{16}$$

and vector

$$\mathbf{b}_k = \left[ \delta g_{\hat{R}}(\hat{x}_p, \hat{y}_q) - G\Delta\rho \sum_{i,j=-(N+M)}^{N+M-1} J_{\Omega_{ij}}(\hat{x}_p, \hat{y}_q, h_{ij}^{(k)}) \right]_{pq} \quad (17)$$

Once vector $\mathbf{h}_k = \left[ h_{ij}^{(k)} \right]_{ij}$ has been obtained, the iterative matrix form of Equation (10) is expressed as

$$A_k\mathbf{h}_{k+1} = \mathbf{b}_k + A_k\mathbf{h}_k \quad (18)$$

As the known data $\delta g_{\hat{R}}(\hat{x}_p, \hat{y}_q)$ in Equations (10) or (18) are given only on $R$, and the sea depths $h_{ij}$ to be solved (where $i, j = -(N+M), \cdots, 0, \cdots, N+M-1$) contain the sea depths of the boundary region in addition to those of $R$, directly solving Equation (18) may lead to poor solvability of the system of equations. To ensure solvability, a regularization factor $\alpha > 0$ is introduced, namely, the actual solved system of equations is expressed as

$$(A_k^T A_k + \alpha E)\mathbf{h}_{k+1} = A_k^T(\mathbf{b}_k + A_k\mathbf{h}_k) \quad (19)$$

where $E$ is the unit matrix. Notably, Equation (19) has a unique solution $\mathbf{h}_{k+1}$ if $\mathbf{h}_k$ is known.

Notably, the sea depth below the boundary sea $\hat{R} - R$ is divergent when iteratively solving Equation (19). To ensure convergence of the iterative process, the sea depth below $\hat{R} - R$ is always considered as the average of the sea depth below $R$ in each iteration. After this treatment, Equation (19) can be iteratively solved. The flow construction of the analytical iterative algorithm is shown in Figure 3.

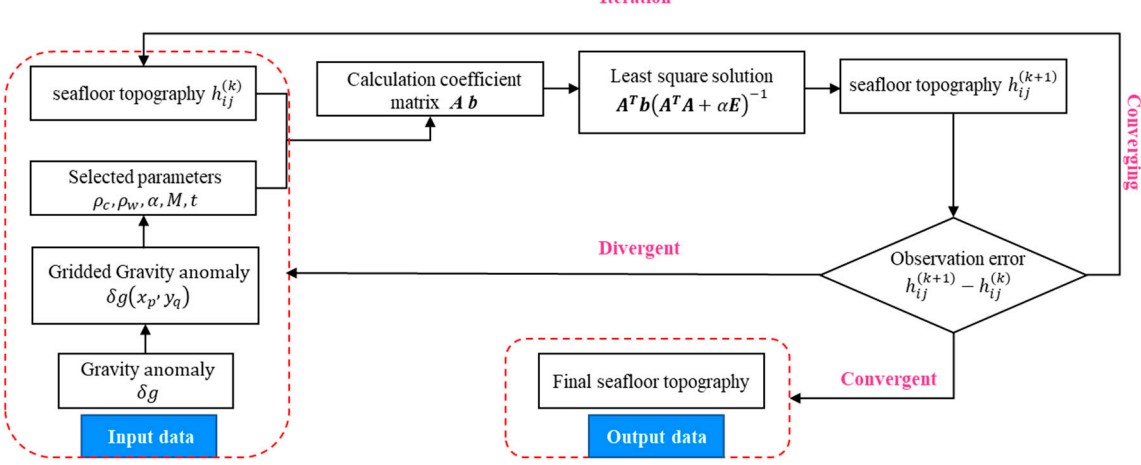

**Figure 3.** Flow chart of the iterative algorithm for the seafloor topography.

## 3. Simulation Experiment

### 3.1. Selection of Some Parameters

This section discusses the solvability of Equation (10) or (19) by simulations, namely, by only considering the boundary effect. In this section, the bedrock and seawater densities are chosen as $\rho_c = 2.7 \times 10^3$ kg/m³ and $\rho_w = 1.03 \times 10^3$ kg/m³, respectively, namely, $\Delta\rho = -1.67 \times 10^3$ kg/m³. Notably, the smaller the step length of the partition for the target region $R$, the higher the accuracy of the solved sea depth beneath $R$. However, as the gravity anomaly on the sea surface in the actual calculation has a resolution of $1' \times 1'$, the step length is always chosen as $t = 2$ km in simulation computations. Additionally, as the boundary effect is considered, the extension number $M$ for $\hat{R}$ should be carefully chosen. According to discussions by Dixon et al. [25] and Yu et al. [34], we choose that $M = 10$, namely, $\hat{R}$ is obtained by extending $R$ outward for 20 km.

We then selected a sea area of 96 km $\times$ 96 km in the South China Sea as $\hat{R}$; its internal sea area of 56 $\times$ 56 km was the target region $R$, and the seafloor topography beneath $\hat{R}$ was chosen from the GEBCO_22 bathymetric model. After gridding $\hat{R}$ by a step length of 2 km,

the seafloor topography beneath $\hat{R}$ is shown in Figure 4a. This implies that the number of partitions for $R$ is $N = 14$. According to the GEBCO_22 model, the maximum undulation of the seafloor topography below $R$ is 610.0 m. Subsequently, this seafloor topography is placed at sea depth $H$ below $\hat{R}$, and the gravity anomaly $\delta g_{\hat{R}}$ generated by it can then be computed. Figure 4b shows the distribution of $\delta g_{\hat{R}}$ on $\hat{R}$ where $H = 6$ km. We aimed to solve the seafloor topography beneath $R$ from $\delta g_{\hat{R}}$ on $R$ using Equation (10) or (19), and then compare it with the "real seafloor topography" beneath $R$. Notably, $H$ is the maximum sea depth of the seafloor topography.

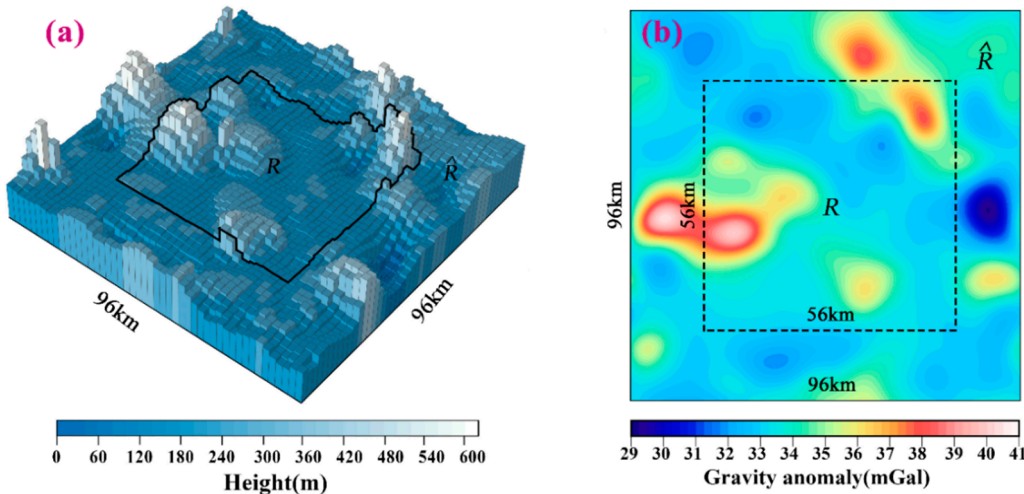

**Figure 4.** (**a**) The 2-km step segmentation seafloor topography beneath the region $\hat{R}$ where the topographic fluctuation is obtained from the GEBCO_22 bathymetry model. (**b**) The distribution of the gravity anomaly on the sea surface generated by this topography when $H = 6$ km.

### 3.2. Selection of Regularization Factors

First, the maximum depth is chosen as $H = 6$ km. For the different regularization parameter $\alpha$ (unit: $10^{-18}\text{s}^{-4}\text{m}^{-2}$), the seafloor topography beneath $R$ is solved using Equation (19) without any error in $\delta g_{\hat{R}}$ and with a random error of 1 mGal in $\delta g_{\hat{R}}$, respectively. Subsequently, compared to the real topography, the root mean square (RMS) error can be computed (Figure 5a). Figure 5a shows that the regularization factor $\alpha$ can be appropriately small if there is no error in $\delta g_{\hat{R}}$ on $R$. For example, when $\alpha = 10^{-5}$, the solved seafloor topography has an error of less than 1.0 m, which is caused by the boundary effect. Additionally, if error exists in $\delta g_{\hat{R}}$ on $R$, the value of $\alpha$ cannot be too small; the reason for this is that the anti-error property of matrix $A_k^T A_k$ is poor. Notably, the eigenvalues of $A_k^T A_k$ corresponding to the sea depths below the boundary region can be easily disturbed, which can lead to a large error in the sea depths below the boundary region, and thus affect the accuracy of the bathymetry below $R$. Therefore, the selection of the regularization factor $\alpha$ must consider the case of error in the gravity anomaly $\delta g_{\hat{R}}$. Figure 5a shows that the optimal value of $\alpha$ should be between 0.1 and 1.0 in the case of maximum depth $H = 6$ km. Simultaneously, the optimal value of $\alpha$ varies with the depth $H$. Generally, the larger the depth $H$, the smaller the optimal value of $\alpha$.

Figure 5b shows the RMS error distributions of the solved seafloor topography beneath $R$ for different maximum depths $H$ in the cases of no error and a random error of 1 mGal in $\delta g_{\hat{R}}$, respectively, where $\alpha = 1$ is fixed. According to the examination rule for accuracy, the error ratio (i.e., ratio of error to the average sea depth) can be usually used as an index. For example, as the topographic relief is 610.0 m, the average sea depth is approximately 4695.0 m when $H = 5$ km and the RMS error is about 40.0 m (yellow curve in Figure 5b); thus, the error ratio is 0.85%. Figure 5b shows that that all the error ratios are less than 1%; thus, choosing the regularization factor $\alpha = 1$ is appropriate.

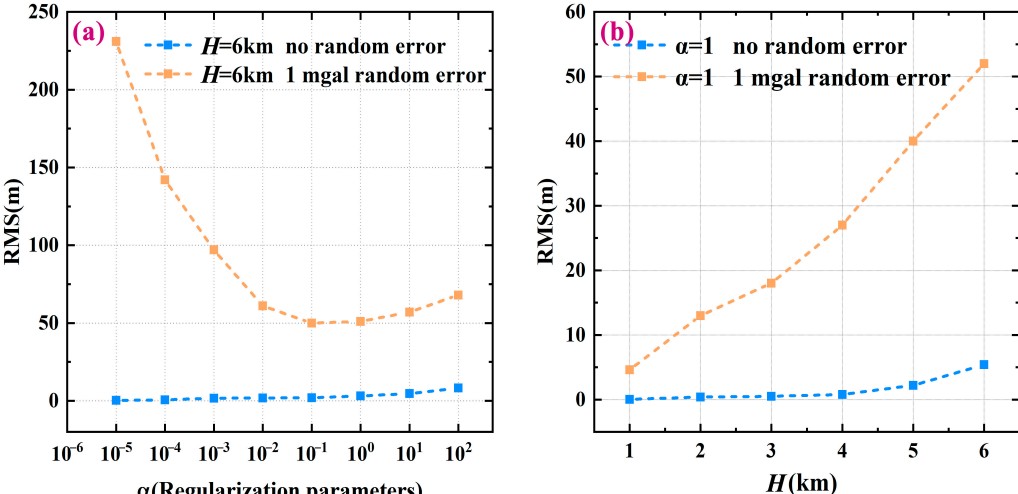

**Figure 5.** (**a**) RMS errors of prediction results corresponding to different $\alpha$ for $H = 6$ km, where the orange and blue dotted curves represent RMS errors for no error and 1 mGal random error, respectively; (**b**) RMS errors of prediction results corresponding to different $H$ for $\alpha = 1$.

### 3.3. Anti-Error Characteristics of the Linearized Systems of Equations

This paragraph discusses the anti-error characteristics of Equation (10) or (19) where $\alpha = 1$ is chosen. As the gravity anomaly on $R$ mainly results from satellite altimetric data, it contains some error. In the following computations, two kinds of errors are added to the gravity anomaly $\delta g_{\hat{R}}$ on $R$: one is the systematic error $\varepsilon$ and the other is the random error with a mean value of zero and standard deviation $\delta$. Subsequently, the seafloor topography beneath $R$ is solved using Equation (19). Furthermore, compared to the real seafloor topography beneath $R$, the RMS errors of the solved seafloor topography can be computed, and their distributions for different maximum depths $H$ are shown in Figure 6a,b, where Figure 6a,b corresponds to the systematic and random errors, respectively. Figure 6a,b shows that: (i) the systematic error in $\delta g_{\hat{R}}$ has less influence on the solved seafloor topography compared to the random error; and (ii) the anti-error ability continuously weakens with increasing sea depth. This is because the deeper the seafloor, the smoother the gravity generated on the sea surface, and the lower its signal-to-noise ratio for the same size of error. For example, for maximum depth $H = 6$ km, the RMS errors of the simulation results for the systematic and random errors are 177.0 m and 221.0 m, respectively, when errors in $\delta g_{\hat{R}}$ are both 5 mGal, indicating that the systematic error has less influence on predicted topography. Additionally, from the statistical results of the random error with an error of 5 mGal in $\delta g_{\hat{R}}$ (Figure 6b), all the error ratios of the solved bathymetries are less than 4%, which fully satisfies the general bathymetry specification necessitating error values of up to 6%. This implies that an accuracy of 5 mGal for the gravity anomaly on the sea surface can guarantee the demand for the inversion of seafloor topography.

Meanwhile, to examine the influence of the initial value $h_{ij}^{(0)}$ and the iterative step in solving Equation (19), the RMS error of the solved seafloor topography using Equation (19) are shown in Figure 7a,b, where no error is added. Figure 7a shows the RMS error convergence curves for different initial values of $h_{ij}^{(0)}$ in the case of $H = 6$ km; notably, the closer the initial value $h_{ij}^{(0)}$ to the true value, the faster the convergence of iterations. Figure 7b shows the relationship between the number of iterations and the RMS error for different maximum sea depths $H$ by considering $h_{ij}^{(0)}$ as 100.0 m. Figure 7b shows that the errors between the solved sea depths and their real values are negligible by solving Equation (19) with five to eight iterations. Overall, we concluded that the sea depth obtained from the iterative scheme, as expressed in Equation (19), rapidly converges to its real value for seafloor topographies with different depths.

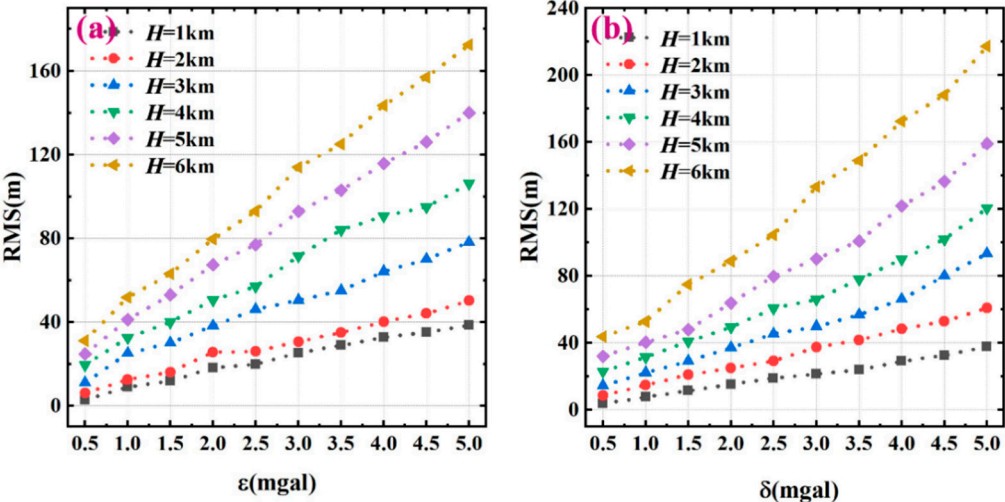

**Figure 6.** (**a**) Anti-error curves of the systematic error for different depths. (**b**) Anti-error curves of random error for different depths.

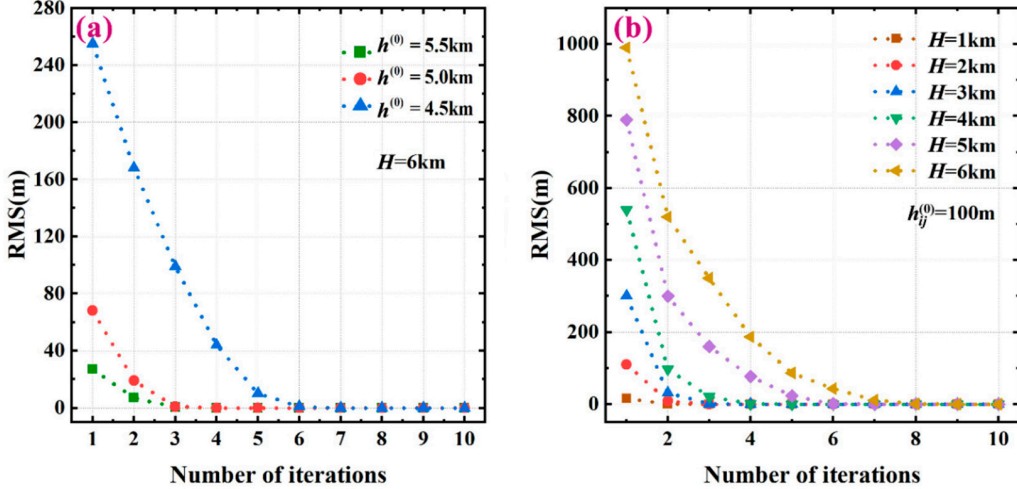

**Figure 7.** (**a**) Iterative processes for different initial values when $H$ = 6 km. (**b**) Iterative processes for different maximum sea depths $H$ with a fixed initial value of 100.0 m.

*3.4. Assessment for the Far Effect*

To examine the far effect and illustrate how to control it by a piecewise bilinear interpolation on $R$, the area $\hat{R}$ shown in Figure 4a is extended to a square area of 200 km² (Figure 8a) and the area $D$ outside $\hat{R}$ can be referred to as the far region. Furthermore, if the seafloor topography beneath $D$ is also given by the GEBCO_22 bathymetry model, the far effect on $R$ can then be obtained by computing the gravity anomaly generated by the seafloor topography beneath $D$ according maximum depth $H$. By choosing maximum depth $H$ = 6 km and assuming $\delta g_D(x, y)$ denotes the far effect on $R$, the difference between $\delta g_D(x, y)$ and $\delta \hat{g}_D(x, y)$ is computed after introducing the piecewise bilinear interpolation function $\delta \hat{g}_D(x, y)$ presented in Section 2.2.3. The statistical results of the difference are shown in Figure 8b. Hence, the error caused by substituting $\delta \hat{g}_D(x, y)$ for $\delta g_D(x, y)$ in Equation (14) is less than 0.3 mGal on average.

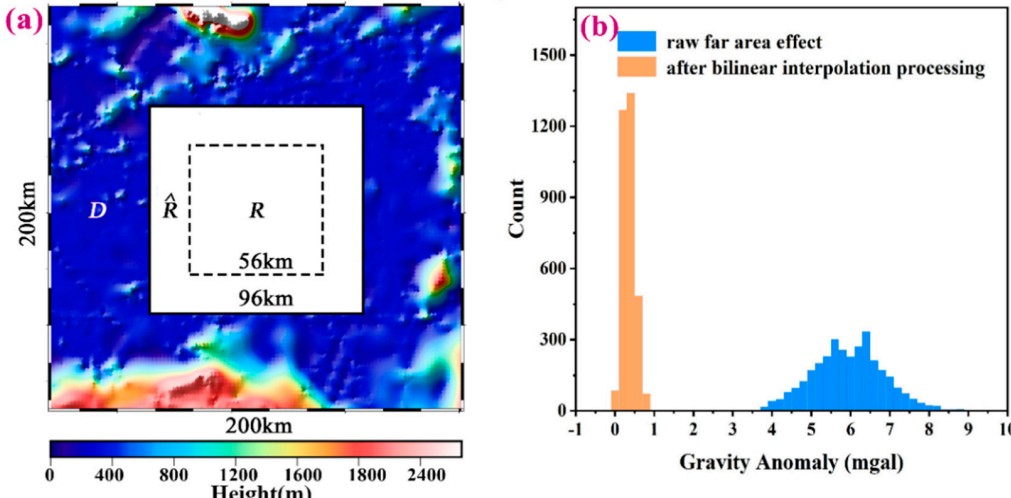

**Figure 8.** (**a**) The simulated topography: $R$, $\hat{R}$ and $D$ represent the corresponding target, boundary, and far regions. (**b**) Histograms of the difference of the far effect and its bilinear interpolation on $R$, where the orange column indicates the distribution of the influence of the gravity anomaly error caused by the far zone after bidirectional interpolation and the blue column represents the original far zone contribution error distribution.

Additionally, assuming that $\delta\overline{g}_D$ is the average value of $\delta g_D(x,y)$ on $R$, the statistical results from the "blue curve" shown in Figure 8b indicate that $\delta g_D(x,y) - \delta\overline{g}_D$ can be approximately referred to as the random error with a standard deviation of 0.8 mGal. Therefore, the term $\delta g_D(x,y)$ in Equation (13) can be also replaced by a constant for simple computation. Notably, Figure 8b is created by choosing $M = 10$ when introducing $\hat{R}$. Thus, if $M$ is larger, the far effect $\delta g_D(x,y)$ on $R$ is closer to its average value $\delta\overline{g}_D$. However, as the condition $M \leq N$ should be satisfied, the choice of $M = 10$ is appropriate in this case.

Although the average value $\delta\overline{g}_D$ is approximately equal to the far effect $\delta g_D(x,y)$, the piecewise bilinear interpolation function $\hat{F}(x,y,\mathbf{F})$ is still recommended owing to the presence of another term (i.e., the deep effect) in Equation (14).

## 4. Actual Application

### 4.1. Target Area and Datasets

A region of the South China Sea at latitudes 13.85°–14.85°N and longitudes 117.25°–118.25°E was selected as the target area $R$ and was then divided into four parts as shown in Figure 9a. The underwater topography of each part is solved from the gravity anomaly using Equation (19), and the whole seafloor topography beneath $R$ can be obtained by splicing four parts together. The advantage of such partition is that the boundary effect can be satisfactorily controlled, thereby weakening the complexity in solving the observation equations.

The gravity anomaly used in this paper is chosen from the DTU17 model [13] and has a resolution of $1' \times 1'$ (Figure 9a); its accuracy is roughly between 1.50 and 5.69 mGal in the South China Sea region [35]. The GEBCO_2022 global topography model published by the International Hydrographic Organization (IHO) is used to evaluate our predicted seafloor topography; its topography under the target area is shown in Figure 9b. Additionally, the data from National Geophysical Data Center (NGDC) with 2512 ship-survey depth points in the target area (Figure 9c) are also used to evaluate our results (www.ngdc.noaa.gov/maps/bathymetry (accessed on 28 September 2022). The GEBCO_2022 global topography model indicates that the maximum and minimum depths in the target area are 4340.0 and 3404.0 m, respectively, and the complexity of the topographic relief is high (Figure 9b); thus, it is appropriate to choose such seafloor topography as the study object.

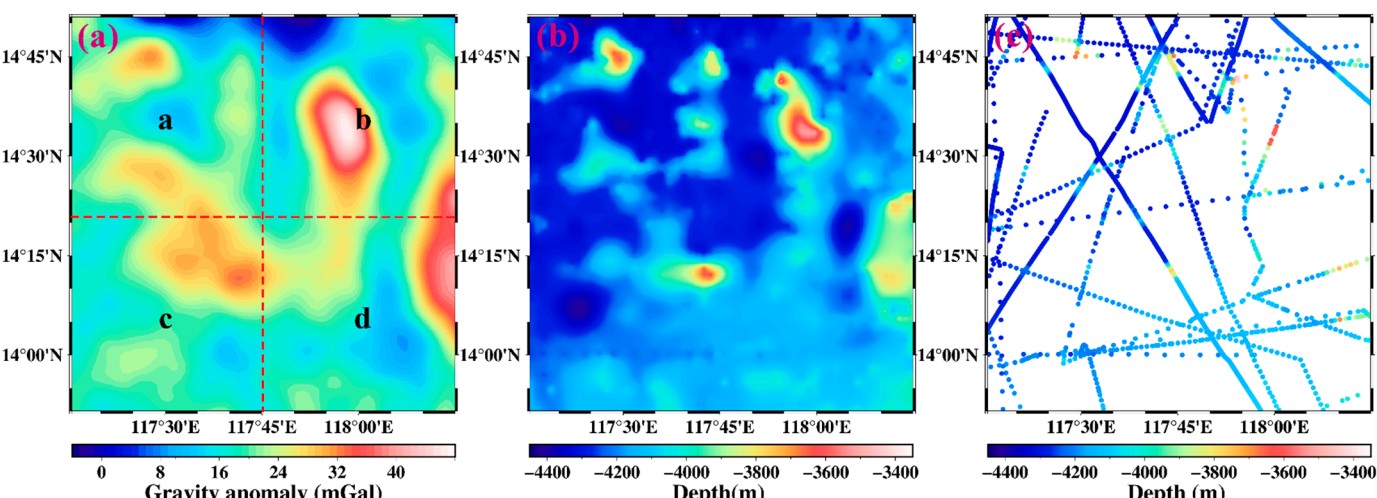

**Figure 9.** (**a**) The distribution of gravity anomaly from the DTU17 model in the target region and its spatial resolution is $1' \times 1'$, where the red dashed line illustrates the zoning so that the areas a, b, c, and d are equally divided. (**b**) The bathymetry from the GEBCO_2022 model in the target region. (**c**) The distribution of the ship soundings data downloaded from the NGDC in the target region.

### 4.2. Results and Comparisons

Based on the algorithm presented in Section 3, the prediction topography is shown in Figure 10, where the regularization parameter $\alpha = 1$, the extension step width $M = 10$ for $\hat{R}$, and the density difference $\Delta\rho = -1.67 \times 10^3$ kg/m$^3$. The comparison between Figures 9a and 10a shows a certain similarity between the gravity anomaly and sea depth, which may indicate the suitability of the GGM method to invert the seafloor topography.

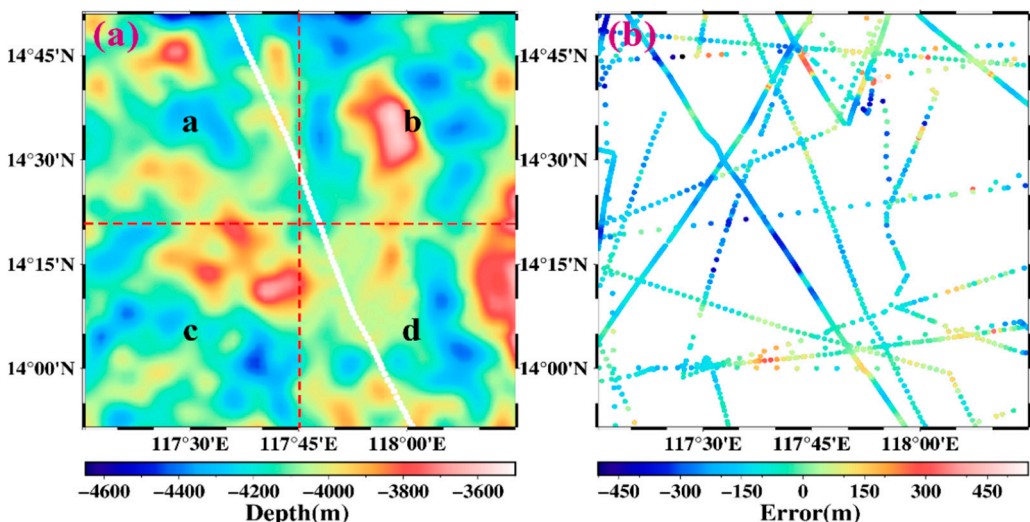

**Figure 10.** (**a**) The prediction topography by the analytic iterative algorithm; the white points are HU939013 ship measurements for subsequent error comparison experiments, where the red dashed line illustrates the zoning, and the a, b, c, d are corresponding to the area divided for calculation in Figure 9a. (**b**) The error distribution of the prediction results compared with the ship soundings.

Then, we analyze the accuracy of the prediction topography. First, compared with the GEBCO_2022 model, the RMS errors of the solved seafloor topographies are listed in the last column of Table 1. Second, compared with the NGDC ship-surveyed depths (Figure 9c), the error distributions are shown in Figure 10b and the main statistical indexes of our result are presented in other columns in Table 1. As the NGDC data is from the ship survey, they

are considered accurate data. The RMS error of our results to the NGDC data is 127.4 m. Hence, the solved topography is acceptable as ship-survey data are not used in our result.

**Table 1.** Main statistical results of the predicted seafloor topography with known data (unit: m).

| Main Indicators | Max Depth | Min Depth | Mean Depth | Max Abs Error | Sys Error | RMS Error | Relative Error | Model Error |
|---|---|---|---|---|---|---|---|---|
| Sub-area a | 4590.9 | 3698.2 | 4063.6 | 480.4 | 25.2 | 140.6 | 3.45% | 148.0 |
| Sub-area b | 4531.4 | 3570.1 | 4018.3 | 533.9 | 17.9 | 116.8 | 2.91% | 134.3 |
| Sub-area c | 4484.8 | 3608.3 | 4011.5 | 437.5 | 22.9 | 144.4 | 3.59% | 153.4 |
| Sub-area d | 4473.0 | 3596.3 | 4007.6 | 426.5 | 13.1 | 107.8 | 2.68% | 110.8 |
| Region *R* | 4590.9 | 3570.1 | 4025.3 | 533.9 | 19.8 | 127.4 | 3.16% | 136.9 |

In this paragraph, a survey line numbered HU939013 (white dashed points in Figure 10a) in the NGDC data is compared with our results. Figure 11a shows the comparison between the gravity anomaly of DTU17 and that along the survey line obtained by forward computation from our predicted seafloor topography, where the maximum absolute, average, and RMS differences are 5.3 mGal, 0.4 mGal, and 2.2 mGal, respectively. Figure 11b shows the comparison of the sea depths, where the maximum absolute, average, and RMS errors of depths along the survey line are 146.3 m, 14.9 m, and 73.42 m. Figure 6b shows the anti-error analysis results; when the maximum depth is 5 km and the random error is 2.5 mGal, the RMS error of the simulation result is 79.0 m. The RMS error of the gravity anomaly difference along the survey line is 2.2 mGal, and the corresponding RMS error for the prediction depth is 73.4 m. This indicates that the numerical simulation results can reflect the final prediction accuracy to a certain extent.

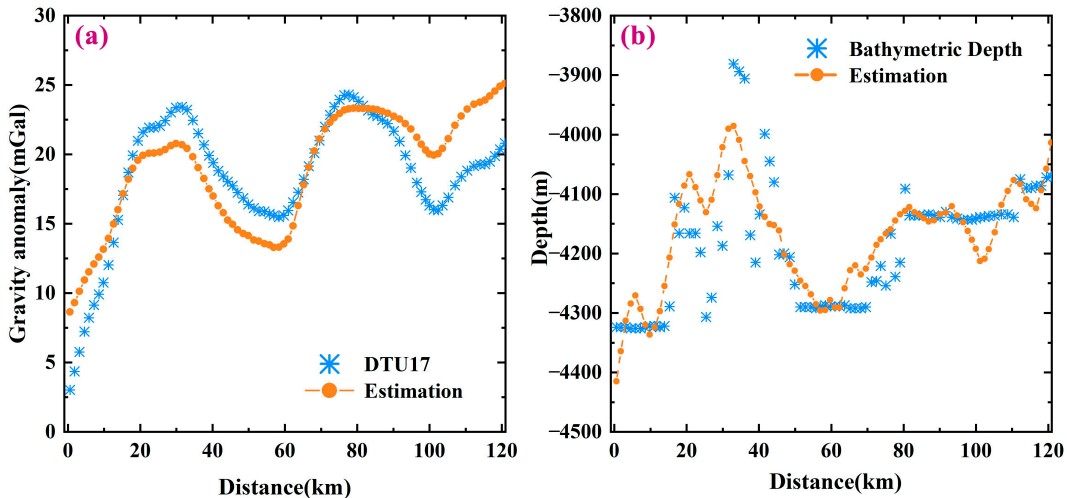

**Figure 11.** (**a**) Comparison between the gravity anomaly of the DTU17 and that obtained in our results by forward computation along the line labeled HU939013. (**b**) Comparison between ship-survey depths and our predicted depths along the line labeled HU939013.

Notably, the known bathymetry data must first be applied to examine the accuracy of the predicted seafloor topography. However, for a certain region on the sea surface, the bathymetry data are mainly obtained along the ship route; thus, its distribution may be relatively sparse in the region. Therefore, using only the bathymetry data as a standard in examining the accuracy of seafloor topography is not comprehensive. Notably, Equation (7) indicated the relationship between the sea depth and gravity anomaly on the sea surface; thus, the gravity anomaly on the sea surface can also be used as an auxiliary standard to evaluate the accuracy of the seafloor topography. Dixon et al. [25] verified that the part of gravity anomaly with wavelengths larger than 30 km is mainly controlled by the far

topography, and only the high frequency part with wavelengths less than 30 km can be used to examine the accuracy of the seafloor topography.

Then, the gravity anomalies on the target region $R$ can be obtained by forward computations for the solved seafloor topography and the corresponding GEBCO_22 topography model, respectively, and their RMS differences to the DTU17 gravity anomaly are computed after subtracting the DTU17 gravity anomaly and filtering out the low-frequency parts with wavelengths larger than 30 km [36]. Notably, such RMS differences can be considered as a match degree with respect to the DTU17; namely, the smaller the RMS difference, the better the matching of the seafloor topography with DTU17. By computations, the RMS differences to the DTU17 on $R$ for the solved topography and GEBCO_22 model are 1.0 mGal and 1.8 mGal, respectively, which implies that our results are a better match with the DTU 17 gravity field model than that obtained by the GEBCO-22 bathymetry model. Therefore, the solved topography is better than one from the GEBCO-22 bathymetry model on $R$ to some extent.

Finally, we indicate that the seafloor topography solved in this paper only uses the gravity anomaly on target region $R$, and does not employ any known ship survey data. Additionally, the measured sea depth data along the ship route can be regarded as a local index to examine the seafloor topography, whereas the matching degree with the gravity anomaly can be regarded as an overall index in the target region.

## 5. Conclusions

In this paper, the grid step length was 2 km, implying that the topography undulations within an area of 2 km $\times$ 2 km were represented by the average depth, which meant that the topography undulations within 2 km $\times$ 2 km could not be identified [21]. Hopefully, the next generation of Surface Water and Ocean Topography (SWOT) satellites may revolutionize the improvement of marine gravity anomalies with a spatial resolution of 1 km [37]. This may significantly improve the prediction accuracy of seafloor topography. Overall, it is important for improving the accuracy of topography prediction to obtain gravity data with higher resolutions and higher accuracies.

The advantages of the analytical iterative method established in this paper are as follows: first, we directly utilized the original gravity anomaly data without filtering or separating the long/short-waves; second, it was not required to introduce the isostatic response function with empirical parameters. The only prerequisite was to weaken the influence of the boundary and far region effects to solve the equations together, which could simplify the calculation.

In summary, we developed a new analytical iterative method to predict topography by building a set of observation equations using the gravity anomaly. Based on numerical simulation experiments, we analyzed the accuracy of the prediction results by refining the error sources and investigating the corresponding error weakening methods. Overall, the main research results of this paper can be summarized as follows: first, based on the gravity expression of a single rectangular prism, we established a system of observation equations between the topography and gravity anomaly, and the solvability of the equations was verified by numerical simulation. Second, the disturbance elements were mainly divided as the boundary, far, and deep effects, and the regularization algorithm and piecewise bilinear interpolation function were used to process the disturbance factors, respectively. Third, the algorithm proposed in this paper was applied to the actual sea area, and ship soundings were used to verify the accuracy of the prediction results. The RMS error of the prediction topography reached 127.4 m in the sea region with an average depth of 4025.3 m, and the relative accuracy of the prediction reached 3.16%.

**Author Contributions:** Derivations for main formulas, designs of computations and review paper, J.Y.; Software, computations for arithmetic examples, analysis for computational results and writing original draft, B.A. and H.X.; Analyses for actual seafloor topographies and review paper, Z.S. and Q.W.; Some computations for imitation arithmetic, B.A. and Y.T. All authors have read and agreed to the published version of the manuscript.

**Funding:** The National Nature Science Funds of China (No: 42274010 and No: 41774089).

**Data Availability Statement:** The single-beam data are provided by National Oceanic and Atmospheric Administration (NOAA) [https://www.ncei.noaa.gov/maps/bathymetry/, accessed on 28 September 2022]. The GEBCO_2022 can be downloaded from [https://www.gebco.net/data_and_products/historical_data_sets/, accessed on 28 September 2022]. The DTU17 is available from [https://ftp.space.dtu.dk/pub/DTU17/1_MIN/, accessed on 28 September 2022].

**Acknowledgments:** The authors thank to DTU and NOAA for supplying actual data.

**Conflicts of Interest:** The authors declare no conflict of interest.

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
