# Peer review of "An Iterative Algorithm for Predicting Seafloor Topography from Gravity Anomalies"

_remotesensing, doi:10.3390/rs15041069_

Round 1

Reviewer 1 Report

This manuscript develops a new analytical iterative method to predict seafloor topography, for the problem of approximate linear relationship predicting topography from gravity anomaly. The analytical observation equation is established, and an iterative solving algorithm is proposed. The detailed analytical observation equations are derived by considering different effects, and the disturbance elements is also discussed. The algorithm is important for predicting seafloor topography. There are still issues that need to be addressed.

1.        Lines 40-41: he “T/X, Jason satellite, and Cryosat satellite”, which altimetry missions are they? Topex/Poseion(T/P), Cryosat(Cryosat-2)?

2.        Line 70: The section is about introduction. It is better that Eq. (1) is present in section 2. Here Partker formula should be analyzed and summarized.

3.        Line 92: Here is vertical gravity. You know, gravitation is a body effect.

4.        Line 104: Geoid is not sea surface.

5.        Line 110: Here Nagy et al -> Nagy et al.

6.        Line 112: Please show the detailed definition of local coordinate system.

7.     Line 150-155, the “where , , and p, q = 0, 0,±1,±2±2N”there are two . Why p,q is ±2N? if considering step t, p,q=0,±1,±2±(2N/t). According the following t=2, p or q  is 0,±1,±2±N in R,  the number of equations in Eq. (10) is (2N/t*2+1)2=(2N+1)2, and the number of unknowns is (2N+2M)2, it cannot ensure the Eq. (10) have a unique solution?

8.     Line 134: What about definitions of boundary, far, and deep regions?

9.     There are several errors for identification symbol, for example Line 179 Line240 , please correct it.

10.  Line 365 “The underwater topography of each part is solved from the gravity anomaly using Eq. (20)”, The Eq. (20) is derived by considering the boundary effect, why is not used the linearization of the Eq. (16) by considering far and deep regions effect?

11.  Line 404, The “Sys error” is what meaning in Tabel 1, and please given the difference between the GEBCO_2022 and ship-surveyed depths.

12.  The iterative algorithm has been derived in detail and verified in local areas. However, what are the advantages of this method compared to conventional methods? (such as GGM or improved GGM). In local complicated topography regions, the vertical gravity gradient is more sensitive to topography at short wavelengths, please give the comparison results derived from the iterative algorithm based on VGG.

13.  References: The format of literatures should be noralized.

Reviewer 2 Report

General comments:

In the proposed manuscript, the Authors touch upon the important topic of obtaining the topography of the seabed. To do this, it is proposed to use some transformation of gravimetric data. The authors also propose an original technique based on the least squares method after linearizing the equations. A comparison of the simulated and obtained bathymetric data collected in the South China Sea shows that the simulation accuracy is 127.4 m, which is about 3.4%. It necessary to say that this topic is very important both from a fundamental scientific point of view and for applied use. But it has a number of pitfalls, and the authors' research is intended to reduce them. Please see my comment below.

1. The overall structure of the manuscript is good. The introduction is generally also good. But the Authors used some mathematical symbols without description. They are explained below in the next section. It is not right. Also, I think this description (from line 63 to the end of the section) should be moved somewhere in the next section. The Introduction section should contain general information and review.

2. Theory and methods. The text is complex, it should be more logical. If Authors describe a model, they must start from description of a coordinate system. Why "A" is a rectangular prism if the bottom is not flat? Moreover, it doesn't matter, because you are integrating A in 3D space (line 99). What is ξ, η and ζ. This should be clear to ordinary readers. What is "O", etc.?

3. Simulation experiment and actual application. These sections are also good. The authors apply the model to real data. I recommend placing the location diagram in Fig. 3 to indicate the area of the experiment over a wider area, including the coastline. It's not important, but it improves the overall impression.

Reviewer 3 Report

The authors designed a new and innovative iterative algorithm for predicting submarine topography from gravity anomalies and compared it with the most commonly used existing model DTU17. It is obvious that by using their method they got better results. Perhaps it would be good if they showed the results of their model in another sample area. I believe that the article does not need to be further corrected and that it can be accepted in this form.
